# Phanerosides A–X, Phenylpropanoid Esters of Sucrose from the Rattans of *Phanera championii* Benth

**DOI:** 10.3390/molecules28124767

**Published:** 2023-06-14

**Authors:** Ya-Jie Hu, Qian Lan, Bao-Jun Su, Dong Liang

**Affiliations:** State Key Laboratory for Chemistry and Molecular Engineering of Medicinal Resources, Collaborative Innovation Center for Guangxi Ethnic Medicine, School of Chemistry and Pharmaceutical Sciences, Guangxi Normal University, Guilin 541004, China

**Keywords:** *Phanera championii* Benth., Fabaceae, phenylpropanoid esters of sucrose, anti-inflammatory, antioxidant

## Abstract

Twenty-four new phenylpropanoid esters of sucrose, phanerosides A–X (**1**–**24**), were isolated from an EtOH extract of the rattans of *Phanera championii* Benth. (Fabaceae). Their structures were elucidated on the basis of comprehensive spectroscopic data analysis. A wide range of structural analogues were presented due to the different numbers and positions of acetyl substituents and the structures of phenylpropanoid moieties. Phenylpropanoid esters of sucrose were isolated from the Fabaceae family for the first time. Biologically, the inhibitory effects of compounds **6** and **21** on NO production in LPS-induced BV-2 microglial cells were better than that of the positive control, with IC_50_ values of 6.7 and 5.2 μM, respectively. The antioxidant activity assay showed that compounds **5**, **15**, **17**, and **24** displayed moderate DPPH radical scavenging activity, with IC_50_ values ranging from 34.9 to 43.9 μM.

## 1. Introduction

Phenylpropanoid esters of sucrose are characterized by the esterification of different hydroxy groups of sucrose with phenylpropanoid moieties [1,2]. These compounds have been isolated mainly from the families Arecaceae, Brassicaceae, Liliaceae, Polygonaceae, Rosaceae, Rutaceae, Smilacaceae, and Sparganiaceae [1,3]. The structural diversity of phenylpropanoid esters of sucrose generates extensive pharmacological antitumor [4,5,6], anti-inflammatory [7,8], antioxidant [9], an anti-HIV [10] activities, which are extremely thrilling to those engaged in medicinal chemistry [11,12].

Fabaceae is the third largest plant family, comprising approximately 800 genera and 20,000 species worldwide [13,14]. As the largest of the several genera in the Fabaceae family, *Phanera* results from the reorganization of *Bauhinia* sensu lato. The genus *Phanera* encompasses about 90–100 species, mainly distributed in tropical Asia and Australasia [15]. As one of them, *Phanera championii* Benth. is widely distributed in Guangxi Province and was first recorded in ‘Nanning Drug Chi’ [16]. It is a well-known folk medicine with a rich history of being used as medication to treat rheumatoid arthritis and epigastric pain [17,18]. Phytochemical investigations of this plant have primarily resulted in the isolation of flavonoids, nitrile glucoside, dibenzofurans, and diterpenoid [16,19,20,21]. In our ongoing research in pursuit of novel and biologically active metabolites from ethnic medicines in Guangxi, 24 new phenylpropanoid esters of sucrose (**1**–**24**) were isolated from the rattans of *P. championii*. Herein, the isolation, structural elucidation, anti-inflammatory, and antioxidant activities in vitro of all the isolated compounds are described.

## 2. Results and Discussion

### Structural Elucidation

Compound **1** (Figure 1) was isolated as a white amorphous powder with a molecular formula of C_30_H_34_O_15_, as determined using HRESIMS (*m*/*z* 657.1798 [M + Na]^+^, calcd for 657.1790) and ^13^C NMR data. The ^1^H NMR spectrum (Table 1) revealed two groups of AA′BB′ aromatic signals [*δ*_H_ 7.49 (2H, d, *J* = 8.8 Hz, H-2″/6″), 6.82 (2H, d, *J* = 8.8 Hz, H-3″/5″) and 7.51 (2H, d, *J* = 8.8 Hz, H-2‴/6‴), 6.81 (2H, d, *J* = 8.8 Hz, H-3‴/5‴)] and two sets of *trans*-olefinic protons [*δ*_H_ 7.72 (1H, d, *J* = 16.0 Hz, H-7″), 6.39 (1H, d, *J* = 16.0 Hz, H-8″) and 7.67 (1H, d, *J* = 16.0 Hz, H-7‴), 6.43 (1H, d, *J* = 16.0 Hz, H-8‴]. In addition, a characteristic doublet with a small coupling constant (*J* = 3.6 Hz) at *δ*_H_ 5.60 was also shown in the ^1^H NMR spectrum, which together with 12 oxygen-bearing carbon signals (including two anomers at *δ*_C_ 105.7 and 90.6) in the ^13^C NMR data (Table 2) supposed the presence of a disaccharide moiety. Furthermore, detailed analysis of the 2D NMR correlations (Figure 2) and chiral HPLC analysis of monosaccharides after acid hydrolysis of **1** revealed that the two sugars were *β*-D-fructose and *α*-D-glucose units connected via C-2→C-1′ to construct a D-sucrose moiety. The ^13^C NMR spectrum showed 30 carbon signals, apart from the 12 carbon signals occupied by the D-sucrose moiety; the other 18 ones were classified as 2 carbonyl carbons (*δ*_C_ 169.2, 168.8) and 16 olefinic or aromatic carbons with the assistance of the HSQC data. Two discrete spin systems, H-7″/H-8″ and H-7‴/H-8‴, in the ^1^H-^1^H COSY spectrum, and the correlations from H-7″ to C-1″/C-2″/C-6″/C-9″ and H-7‴ to C-1‴/C-2‴/C-6‴/C-9‴ in the HMBC spectrum (Figure 2), established two *trans*-*p*-coumaroyl moieties. Subsequently, the key HMBC correlations from H-2′ to C-9″ as well as from H_2_-6′ to C-9‴ suggested that the two *trans*-*p*-coumaroyl moieties were located at C-2′ and C-6′. Thus, the structure of **1** was identified and named phaneroside A.

Compounds **2**–**24** were determined to be structurally related to **1** according to their extremely similar physicochemical properties and NMR data. The latter all showed the characteristics of a core D-sucrose unit, while the variation of substituents and their positions obtained different structures. The structural elucidation of **2**–**24** was as follows.

The molecular formula of compound **2** was assigned as C_30_H_34_O_14_ based on its HRESIMS (*m/z* 641.1847 [M + Na]^+^, calcd for 641.1841) and ^13^C NMR data, with 16 mass units less than **1**. The analogous ^1^H NMR data (Table 1) of **1** and **2** indicated that they were structural analogues, excepting of the presence of a monosubstituted aromatic moiety [*δ*_H_ 7.65 (1H, m, H-2‴/6‴), 7.41 (3H, overlapped, H-3‴/4‴/5‴)] and the absence of an AA′BB′ aromatic unit in **2**. The long-range correlations from phenyl protons to olefinic carbons and correlations from olefinic protons to ester carbonyl carbons in the HMBC spectrum indicated that a *trans*-*p*-coumaroyl group and a *trans*-cinnamoyl group were presented (Figure 3). The HMBC correlations from H-2′ (*δ*_H_ 4.73) to C-9″ and H_2_-6′ (*δ*_H_ 4.56, 4.34) to C-9‴ confirmed the structure of **2** as shown, and this compound was named phaneroside B.

Compounds **3** and **4** had the same molecular formula, C_31_H_36_O_16_, as determined using the HRESIMS peaks at *m/z* 687.1895 and 687.1896 ([M + Na]^+^, calcd for 687.1896), respectively, indicating 30 mass units more than **1**, ascribed to a methoxy group. The ^1^H NMR data (Table 1) of **3** and **4** were similar to those of **1**, except for the presence of an ABX aromatic moiety [*δ*_H_ 7.21 (1H, d, *J* = 2.0 Hz, H-2″), 7.11 (1H, dd, *J* = 8.0, 2.0 Hz, H-6″), 6.82 (1H, d, *J* = 8.0 Hz, H-5″) in **3** and 7.25 (1H, br s, H-2‴), 7.10 (1H, br d, *J* = 8.4 Hz, H-6‴), 6.81 (1H, d, *J* = 8.4 Hz, H-5‴) in **4**] and a methoxy [*δ*_H_ 3.90 (3H, s) in **3** and 3.89 (3H, s) in **4**] and the absence of an AA′BB′ aromatic unit. The HMBC correlations from H-5″/3″-OMe to C-3″ in **3** and from H-5‴/3‴-OMe to C-3‴ in **4**, together with the ^1^H-^1^H COSY and HMBC correlations as mentioned previously, suggested that there were a *trans*-feruloyl group and a *trans*-*p*-coumaroyl group in both **3** and **4** (Figure 3). The complete structures of **3** and **4** were further established by the HMBC correlations from the protons of the sucrose unit to carbonyl carbons. Therefore, compounds **3** and **4** were determined to be phanerosides C and D, respectively.

Compound **5** gave the molecular formula of C_32_H_38_O_17_, as determined by an HRESIMS ion at *m/z* 717.1990 ([M + Na]^+^, calcd for 717.2001). The NMR data (Table 1 and Table 2) indicated the presence of two *trans*-feruloyl moieties and a D-sucrose unit. The key HMBC cross-peaks from H-2′ (*δ*_H_ 4.73) to C-9″ and H_2_-6′ (*δ*_H_ 4.54, 4.31) to C-9‴ suggested that the two *trans*-feruloyl moieties were linked to C-2′ and C-6′ of glucopyranose unit (Figure 3). Thus, the structure of compound **5** was identified and named phaneroside E.

Compounds **6**–**8** shared the same molecular formula of C_32_H_36_O_16_, as obtained from their respective HRESIMS data. Their molecular masses were 42 mass units more than that of **1**, which combined with the 1D NMR data of **6**–**8** (Table 3 and Table 5) suggested that an acetyl group [*δ*_H_ 2.01 (3H, s), *δ*_C_ 172.0, 20.6 in **6**; *δ*_H_ 2.18 (3H, s), *δ*_C_ 172.2, 20.8 in **7**; *δ*_H_ 2.05 (3H, s), *δ*_C_ 172.4, 20.8 in **8**] existed in each compound. The key HMBC correlations from H_2_-1 (*δ*_H_ 4.11, 4.06) to carbonyl (*δ*_C_ 172.0) in **6**, from H-3 (*δ*_H_ 5.43) to carbonyl (*δ*_C_ 172.2) in **7**, and from H-4 (*δ*_H_ 5.30) to carbonyl (*δ*_C_ 172.4) in **8** indicated that the acetyl group was attached to C-1, C-3, and C-4 of the fructofuranose unit in compounds **6**, **7**, and **8**, respectively (Figure 4). In addition, the chemical shifts of the protons linked to the acetyl group in **6**–**8** were obviously shifted downfield compared to those of **1**, which also supported the above description. Thus, the structures of **6**, **7**, and **8** were established and named phanerosides F, G, and H, respectively.

Compounds **9**–**11** were assigned the same molecular formula, C_33_H_38_O_17_, according to their HRESIMS and ^13^C NMR data. Compounds **9**–**11** were determined to be acetylated derivatives of **4**, as their molecular masses were 42 mass units more than that of **4**. Their ^1^H and ^13^C NMR data (Table 3 and Table 5) showed the characteristics of the acetyl group [*δ*_H_ 2.01 (3H, s), *δ*_C_ 172.0, 20.6 in **9**; *δ*_H_ 2.18 (3H, s), *δ*_C_ 172.2, 20.8 in **10**; *δ*_H_ 2.04 (3H, s), *δ*_C_ 172.4, 20.8 in **11**]. Furthermore, the key HMBC correlations from H_2_-1 (*δ*_H_ 4.11, 4.06) to carbonyl (*δ*_C_ 172.0) in **9**, from H-3 (*δ*_H_ 5.44) to carbonyl (*δ*_C_ 172.2) in **10**, and from H-4 (*δ*_H_ 5.28) to carbonyl (*δ*_C_ 172.4) in **11** located the acetyl group at C-1, C-3, and C-4 for compounds **9**, **10**, and **11**, respectively. Thus, the structures of compounds **9**–**11** (phanerosides I–K) were defined as shown.

Compounds **12**–**14** possessed identical molecular formula, C_33_H_38_O_17_, as determined by their respective HRESIMS ion peaks at *m*/*z* 729.2018, 729.2010, and 729.1990 ([M + Na]^+^, calcd for 729.2001), which were 42 mass units more than that of **3**. Detailed analysis of their 1D NMR data (Table 4 and Table 5) indicated that compounds **12**–**14** were acetylated derivatives of **3**, with a difference in the position of the acetyl group. In their HMBC spectra, the correlations from H_2_-1 (*δ*_H_ 4.14, 4.03) to carbonyl (*δ*_C_ 172.0) in **12**, from H-3 (*δ*_H_ 5.43) to carbonyl (*δ*_C_ 172.2) in **13**, and from H-4 (*δ*_H_ 5.31) to carbonyl (*δ*_C_ 172.3) in **14** suggested that the acetyl group was linked to C-1, C-3, and C-4 in **12**, **13**, and **14**, respectively. Therefore, the structures of compounds **12**–**14** (phanerosides L–N) were established as shown.

Compounds **15**–**17** were determined to have the same molecular formula, C_34_H_40_O_18_, on the basis of their HRESIMS and ^13^C NMR data, displaying 42 mass units more than that of **5**. Compounds **15**–**17** were suggested to be acetylated derivatives of **5** after an analysis of their 1D NMR data (Table 4 and Table 5). The acetyl group was, respectively, located at C-1, C-3, and C-4 of the fructofuranose unit in **15**, **16**, and **17**, which was confirmed using the key HMBC correlations from the protons of the sugar unit to corresponding carbonyl carbons. Accordingly, the structure of compounds **15**–**17** (phanerosides O–Q) were determined as shown.

Compounds **18** and **19** had the same molecular formula, C_34_H_40_O_18_, as **17** according to their HRESIMS and ^13^C NMR data. The 1D NMR data (Table 4 and Table 5) closely resembled those of **17**, with the exception that one of the two *trans*-feruloyl groups was replaced by a *cis*-feruloyl group according to the smaller coupling constants of ^3^*J*_7″,8″_ (12.6 Hz) in **18** and ^3^*J*_7‴,8‴_ (13.2 Hz) in **19**. The *cis*-feruloyl group was linked to C-2′ and C-6′ of the glucopyranose unit in **18** and **19**, respectively, which was proved by the key HMBC cross-peaks from H-2′/H-7″ to C-9″ in **18** and from H-6′/H-7‴ to C-9‴ in **19** (Figure 5). Thus, the structures of compounds **18** and **19** were identified and named phanerosides R and S, respectively.

Compounds **20** and **21** were suggested to have the same molecular formula, C_34_H_38_O_17_, owing to their coincident positive HRESIMS ion at *m/z* 741.2001 [M + Na]^+^ (calcd for 741.2001), indicating 42 mass units more than that of **7**. Comparison of their 1D NMR data (Table 6 and Table 7) with those of **7** disclosed that one more acetyl group was presented in both **20** and **21**. The key HMBC correlations from H_2_-1 (*δ*_H_ 4.17, 4.00) to carbonyl (*δ*_C_ 172.0) and H-3 (*δ*_H_ 5.29) to carbonyl (*δ*_C_ 172.1) in **20**, and from H-3 (*δ*_H_ 5.65) to carbonyl (*δ*_C_ 171.8) and H-4 (*δ*_H_ 5.48) to carbonyl (*δ*_C_ 172.0) in **21**, indicated that the two acetyl groups were located at C-1 and C-3 in **20** and C-3 and C-4 in **21** (Figure 6). Thus, the structures of compounds **20** and **21** were characterized and named phanerosides T and U, respectively.

Compounds **22** and **23** showed the same molecular formula, C_35_H_40_O_18_, as determined by their ^13^C NMR data and respective HRESIMS ion peaks at *m*/*z* 771.2093 and 771.2112 ([M + Na]^+^, calcd for 771.2107). Analysis of the NMR data (Appendix A) of **22** and **23** proclaimed that the sugar moieties in both compounds were acylated by a *trans*-ferulic acid, a *trans*-*p*-coumaric acid, and two acetic acids. The *trans*-feruloyl and *trans*-*p*-coumaroyl units were, respectively, located at C-2′ and C-6′ in **22** based on the key HMBC correlations from H-2′ to C-9″ and H_2_-6′ to C-9‴, while the locations of these two substituents in **23** were the opposite. The key HMBC correlations from H-3 (*δ*_H_ 5.65) to carbonyl (*δ*_C_ 172.0) and H-4 (*δ*_H_ 5.50) to carbonyl (*δ*_C_ 171.8) in **22**, and from H_2_-1 (*δ*_H_ 4.16, 3.99) to carbonyl (*δ*_C_ 171.9) and H-3 (*δ*_H_ 5.30) to carbonyl (*δ*_C_ 172.1) in **23**, indicated that the two acetyl groups were placed at C-3 and C-4 in **22** and C-1 and C-3 in **23**. Accordingly, the structures of compounds **22** and **23** (phanerosides V and W) were defined as shown.

The molecular formula of compound **24** was assigned as C_36_H_42_O_19_ based on the HRESIMS (*m/z* 801.2226 [M + Na]^+^, calcd for 801.2213) and ^13^C NMR data. The NMR data (Appendix A) indicated that it possessed two *trans*-feruloyl moieties, D-sucrose, and two acetyl groups. The two *trans*-feruloyl moieties were linked to C-2′ and C-6′ of the glucopyranose ring, and two acetyl groups were attached to C-3 and C-4 of the fructofuranose ring, according to the key HMBC correlations as shown in Figure 6. Thus, the structure of compound **24** was identified and named phaneroside X.

In Vitro Anti-inflammatory Effects of Compounds **1**–**24**

Nitric oxide (NO) is one of the major inflammatory mediators, and phenylpropanoid esters of sucrose have been previously reported to possess potent anti-inflammatory activity [3,7,22]. Therefore, all the isolates were evaluated in vitro for their anti-inflammatory potential via the Griess reaction in LPS-induced BV-2 microglial cells (Figure 7) [23]. Especially, compounds **6** and **21** exhibited potent inhibitory activities on NO production, with IC_50_ values of 6.7 ± 1.7 and 5.2 ± 3.5 μM, which were better than the positive control, L-NMMA (IC_50_ = 7.0 ± 2.7 μM). Compounds **10**, **14**, and **19** showed moderate inhibitory effects on NO production, with IC_50_ values of 72.7, 46.0, and 57.7 μM, respectively. These results suggested that the anti-inflammatory activities of these compounds were not determined by a single variable, while the type, number, and position of the substituents may all affect their inhibitory activities.

b.Antioxidant Effects of Compounds **1**–**24**

Many isolated phenylpropanoid esters of sucrose are thought to act as potential antioxidants [3]. Consequently, their antioxidant activities were also tested using the DPPH radical scavenging assay [24]. Compounds **5**, **15**, **17**, and **24** exhibited moderate inhibitory effects with EC_50_ values of 43.9 ± 0.2, 43.8 ± 0.1, 34.9 ± 0.1, 39.4 ± 0.3 μM, respectively. As it stands, the compounds whose C-2′ and C-6′ of the glucopyranose ring were both substituted by *trans*-feruloyl groups showed a more positive impact on their antioxidant effects.

## 3. Materials and Methods

### General Experimental Procedures

Optical rotations were obtained on a JASCO P-2000 polarimeter. UV absorption spectra were determined on a PerkinElmer 650 spectrophotometer. NMR spectra were acquired on a 400 or 600 MHz Bruker AVANCE apparatus. Chemical shifts are expressed in *δ* (ppm) and referenced to the solvent residual peak. HRESIMS data were obtained on an Agilent 6545 Q-TOF LC-MS spectrometer. The other instruments and materials serving for the isolation and purification of compounds were coincident with previous papers [25,26].

Plant Material

The rattans of *P. championii* Benth. were collected in November 2020 in Guilin, Guangxi Province, People’s Republic of China (GPS: 24°47′32.7″ N 110°27′36.8″ E). The specimen (No. PC-202011) was authenticated by Professor Shao-Qing Tang (College of Life Science, Guangxi Normal University) and deposited at the State Key Laboratory for Chemistry and Molecular Engineering of Medicinal Resources, Guangxi Normal University.

b.Extraction and Isolation

The dried rattans of *P. championii* (21.0 kg) were soaked for 12 h in 95% aqueous EtOH (100 L) at room temperature, and then extracted three times with 95% aqueous EtOH (3 × 100 L) via refluxing. The filtrate was concentrated under reduced pressure to afford 4.5 kg of crude extract, which was suspended in H_2_O and successively partitioned with EtOAc and *n*-BuOH. The EtOAc partition (1.8 kg) was separated via silica gel (200–300 mesh) column chromatography (CC), eluting with a gradient of CH_2_Cl_2_/MeOH (from 1:0 to 1:1) to give 11 fractions (Frs.1–11).

Fr.6 (20.7 g) was separated via C_18_ reversed-phase (RP) CC, eluting with a gradient of MeOH-H_2_O (30:70 to 70:30) to give 11 subfractions (Frs.6.1–6.11).

Fr.6.2 (656.8 mg) was subjected to Sephadex LH-20 CC (MeOH) to yield six subfractions (Frs.6.2.1–6.2.6). Fr.6.2.5 (183.9 mg) was applied to silica gel (200–300 mesh) CC, eluting with CH_2_Cl_2_/MeOH (50:1 to 5:1) to obtain nine subfractions (Frs.6.2.5.1–6.2.5.9). Fr.6.2.5.8 (14.3 mg) was further purified using semipreparative RP-HPLC (CH_3_CN/H_2_O, 22:78, 8.0 mL/min) to afford compound **5** (10.0 mg, *t*_R_ 37.1 min). Compounds **3** (12.0 mg, *t*_R_ 38.5 min) and **4** (35.0 mg, *t*_R_ 42.4 min) were obtained using semipreparative RP-HPLC (CH_3_CN/H_2_O, 22:78, 8.0 mL/min) from Fr.6.2.5.9 (91.5 mg).

Fr.6.3 (475.2 mg) was separated via a Sephadex LH-20 column, eluting with MeOH to yield six subfractions (Frs.6.3.1–6.3.6). Fr.6.3.5 (223.1 mg) was fractionated using silica gel (200–300 mesh) with gradient elution (CH_2_Cl_2_/MeOH, 50:1 to 5:1) to provide seven subfractions (Frs.6.3.5.1–6.3.5.7). Purification of Fr.6.3.5.3 (25.4 mg) using semipreparative RP-HPLC (CH_3_CN/H_2_O, 24:76, 8.0 mL/min) to yield compounds **18** (4.0 mg, *t*_R_ 54.9 min), **17** (12.0 mg, *t*_R_ 64.4 min), and **19** (3.0 mg, *t*_R_ 77.7 min). Fr.6.3.5.4 (93.5 mg) was purified using semipreparative RP-HPLC (CH_3_CN/H_2_O, 24:76, 8.0 mL/min) to obtain compounds **14** (15.0 mg, *t*_R_ 48.1 min), **16** (3.0 mg, *t*_R_ 49.2 min), and **11** (25.0 mg, *t*_R_ 52.0 min). Fr.6.3.5.5 (32.6 mg) was further purified using semipreparative RP-HPLC (CH_3_CN/H_2_O, 24:76, 8.0 mL/min) to afford compounds **13** (8.0 mg, *t*_R_ 40.2 min) and **10** (18.5 mg, *t*_R_ 44.4 min).

Fr.6.5 (532.6 mg) was separated via Sephadex LH-20 CC (MeOH) and then silica gel (200–300 mesh) CC (CH_2_Cl_2_/MeOH, 50:1 to 8:1) to provide nine subfractions (Frs.6.5.1–6.5.9). Fr.6.5.3 (15.0 mg) was further purified using semipreparative RP-HPLC (CH_3_CN/H_2_O, 25:75, 8.0 mL/min) to produce compound **24** (6.0 mg, *t*_R_ 77.7 min). Fr.6.5.4 (59.0 mg) was chromatographed on a Sephadex LH-20 column using MeOH as a solvent and then purified using semipreparative RP-HPLC (CH_3_CN/H_2_O, 27:73, 8.0 mL/min) to give compound **22** (3.5 mg, *t*_R_ 89.4 min). Fr.6.5.5 (35.4 mg) and Fr.6.5.9 (11.9 mg) were further purified using semipreparative RP-HPLC (CH_3_CN/H_2_O, 27:73, 8.0 mL/min) to produce compounds **23** (11.0 mg, *t*_R_ 50.0 min) and **2** (3.0 mg, *t*_R_ 58.1 min), respectively. Further purification of Fr.6.5.6 (26.5 mg) using semipreparative RP-HPLC (CH_3_CN/H_2_O, 26:74, 8.0 mL/min) yielded compound **15** (7.0 mg, *t*_R_ 42.9 min). Purification of Fr.6.5.7 (26.5 mg) using semipreparative RP-HPLC (CH_3_CN/H_2_O, 25:75, 8.0 mL/min) gave compounds **12** (6.0 mg, *t*_R_ 42.3 min) and **9** (11.0 mg, *t*_R_ 45.0 min).

Fr.7 (14.0 g) was chromatographed over an MCI-gel column, eluting with MeOH/H_2_O (20:80 to 85:15) to give 12 subfractions (Frs.7.1–7.12).

Fr.7.4 (527.6 mg) was fractionated using a silica gel (200–300 mesh) column (CH_2_Cl_2_/MeOH, 50:1 to 5:1) to obtain seven subfractions (Frs.7.4.1–7.4.7). Fr.7.4.6 (252.4 mg) was purified using semipreparative RP-HPLC (CH_3_CN/H_2_O, 20:80, 8.0 mL/min) to give compound **1** (80.0 mg, *t*_R_ 49.1 min).

Fr.7.6 (740.5 mg) was partitioned into six subfractions (Frs.7.6.1–7.6.6) using a Sephadex LH-20 column (MeOH). Fr.7.6.4 (449.3 mg) was separated via silica gel (200–300 mesh) CC to provide eight subfractions (Frs.7.6.4.1–7.6.4.8). Fr.7.6.4.4 (77.3 mg) was further purified using semipreparative RP-HPLC (CH_3_CN/H_2_O, 24:76, 8.0 mL/min) to yield compound **8** (25.0 mg, *t*_R_ 39.8 min). Fr.7.6.4.6 (80.6 mg) was purified using semipreparative RP-HPLC (CH_3_CN/H_2_O, 24:76, 8.0 mL/min) to obtain compounds **7** (30.0 mg, *t*_R_ 44.1 min) and **6** (8.0 mg, *t*_R_ 56.2 min).

Fr.7.7 (833.7 mg) was separated using Sephadex LH-20 (MeOH) to obtain eight subfractions (Frs.7.7.1–7.7.8). Fr.7.7.2 (258.5 mg) was then fractionated via silica gel (200–300 mesh) CC to give six subfractions (Frs.7.7.2.1–7.7.2.6). Fr.7.7.2.2 (74.3 mg) and Fr.7.7.2.3 (69.8 mg) were purified using semipreparative RP-HPLC (CH_3_CN/H_2_O, 27:73, 8.0 mL/min) to yield compounds **21** (25.0 mg, *t*_R_ 54.4 min) and **20** (35.0 mg, *t*_R_ 50.2 min), respectively.

c.Physicochemical Properties and Spectroscopic Data of Compounds **1**–**24**

Phaneroside A (**1**): white amorphous powder; [α]${}_{D}^{20}$ + 33 (*c* 0.06, MeOH); UV (MeOH) *λ*_max_ (log *ε*) 211 (3.87), 228 (3.92), 314 (4.27) nm; IR (KBr) *ν*_max_ 3348, 1696, 1605, 1516, 1171, 1054 cm^−1^; (+) HRESIMS *m/z* 657.1798 [M + Na]^+^ (calcd for C_30_H_34_O_15_Na, 657.1790); ^1^H and ^13^C NMR data, see Table 1 and Table 2. All significant data are presented in Appendix A.

Phaneroside B (**2**): white amorphous powder; [α]${}_{D}^{20}$ + 36 (*c* 0.08, MeOH); UV (MeOH) *λ*_max_ (log *ε*) 217 (3.95), 320 (4.25) nm; IR (KBr) *ν*_max_ 3432, 1696, 1605, 1516, 1271, 1169, 1050 cm^−1^; (+) HRESIMS *m/z* 641.1847 [M + Na]^+^ (calcd for C_30_H_34_O_14_Na, 641.1841); ^1^H and ^13^C NMR data, see Table 1 and Table 2. All significant data are presented in Appendix A.

Phaneroside C (**3**): white amorphous powder; [α]${}_{D}^{20}$ + 22 (*c* 0.06, MeOH); UV (MeOH) *λ*_max_ (log *ε*) 219 (3.82), 320 (4.10) nm; IR (KBr) *ν*_max_ 3417, 1691, 1631, 1605, 1516, 1170, 1052 cm^−1^; (+) HRESIMS *m/z* 687.1895 [M + Na]^+^, (calcd for C_31_H_36_O_16_Na, 687.1896); ^1^H and ^13^C NMR data, see Table 1 and Table 2. All significant data are presented in Appendix A.

Phaneroside D (**4**): white amorphous powder; [α]${}_{D}^{20}$ + 19 (*c* 0.06, MeOH); UV (MeOH) *λ*_max_ (log *ε*) 231 (3.94), 320 (4.25) nm; IR (KBr) *ν*_max_ 3348, 1694, 1632, 1604, 1516, 1270, 1170 cm^−1^; (+) HRESIMS *m/z* 687.1896 [M + Na]^+^ (calcd for C_31_H_36_O_16_Na, 687.1896); ^1^H and ^13^C NMR data, see Table 1 and Table 2. All significant data are presented in Appendix A.

Phaneroside E (**5**): white amorphous powder; [α]${}_{D}^{20}$ + 36 (*c* 0.06, MeOH); UV (MeOH) *λ*_max_ (log *ε*) 217 (4.07), 237 (4.01), 328 (4.28) nm; IR (KBr) *ν*_max_ 3418, 1694, 1606, 1517, 1246, 1170, 1053 cm^−1^; (+) HRESIMS *m/z* 717.1990 [M + Na]^+^ (calcd for C_32_H_38_O_17_Na, 717.2001); ^1^H and ^13^C NMR data, see Table 1 and Table 2. All significant data are presented in Appendix A.

Phaneroside F (**6**): white amorphous powder; [α]${}_{D}^{20}$ + 31 (*c* 0.06, MeOH); UV (MeOH) *λ*_max_ (log *ε*) 211 (3.86), 228 (3.91), 314 (4.26) nm; IR (KBr) *ν*_max_ 3431, 1693, 1605, 1516, 1171, 1051 cm^−1^; (+) HRESIMS *m/z* 699.1915 [M + Na]^+^ (calcd for C_32_H_36_O_16_Na, 699.1896); ^1^H and ^13^C NMR data, see Table 3 and Table 5. All significant data are presented in Appendix A.

Phaneroside G (**7**): white amorphous powder; [α]${}_{D}^{20}$ + 36 (*c* 0.06, MeOH); UV (MeOH) *λ*_max_ (log *ε*) 211 (3.88), 228 (3.93), 315 (4.28) nm; IR (KBr) *ν*_max_ 3421, 1694, 1606, 1516, 1259, 1171, 1059 cm^−1^; (+) HRESIMS *m/z* 699.1898 [M + Na]^+^ (calcd for C_32_H_36_O_16_Na, 699.1896); ^1^H and ^13^C NMR data, see Table 3 and Table 5. All significant data are presented in Appendix A.

Phaneroside H (**8***)*: white amorphous powder; [α]${}_{D}^{20}$ + 33 (*c* 0.06, MeOH); UV (MeOH) *λ*_max_ (log *ε*) 211 (3.93), 228 (3.98), 314 (4.32) nm; IR (KBr) *ν*_max_ 3326, 1696, 1605, 1516, 1171, 1063 cm^−1^; (+) HRESIMS *m/z* 699.1899 [M + Na]^+^ (calcd for C_32_H_36_O_16_Na, 699.1896); ^1^H and ^13^C NMR data, see Table 3 and Table 5. All significant data are presented in Appendix A.

Phaneroside I (**9**): white amorphous powder; [α]${}_{D}^{20}$ + 23 (*c* 0.06, MeOH); UV (MeOH) *λ*_max_ (log *ε*) 217 (4.00), 319 (4.29) nm; IR (KBr) *ν*_max_ 3427, 1695, 1632, 1605, 1516, 1270, 1170, 1051 cm^−1^; (+) HRESIMS *m/z* 729.2015 [M + Na]^+^ (calcd for C_33_H_38_O_17_Na, 729.2001); ^1^H and ^13^C NMR data, see Table 3 and Table 5. All significant data are presented in Appendix A.

Phaneroside J (**10**): white amorphous powder; [α]${}_{D}^{20}$ + 23 (*c* 0.06, MeOH); UV (MeOH) *λ*_max_ (log *ε*) 216 (3.98), 320 (4.28) nm; IR (KBr) *ν*_max_ 3436, 1691, 1632, 1605, 1516, 1268, 1170 cm^−1^; (+) HRESIMS *m/z* 729.2012 [M + Na]^+^ (calcd for C_33_H_38_O_17_Na, 729.2001); ^1^H and ^13^C NMR data, see Table 3 and Table 5. All significant data are presented in Appendix A.

Phaneroside K (**11**): white amorphous powder; [α]${}_{D}^{20}$ + 19 (*c* 0.05, MeOH); UV (MeOH) *λ*_max_ (log *ε*) 216 (4.01), 319 (4.29) nm; IR (KBr) *ν*_max_ 3429, 1695, 1632, 1605, 1516, 1268, 1170, 1053 cm^−1^; (+) HRESIMS *m/z* 729.1989 [M + Na]^+^ (calcd for C_33_H_38_O_17_Na, 729.2001); ^1^H and ^13^C NMR data, see Table 3 and Table 5. All significant data are presented in Appendix A.

Phaneroside L (**12**): white amorphous powder; [α]${}_{D}^{20}$ + 19 (*c* 0.06, MeOH); UV (MeOH) *λ*_max_ (log *ε*) 218 (4.03), 319 (4.30) nm; IR (KBr) *ν*_max_ 3436, 1695, 1632, 1605, 1516, 1274, 1171 cm^−1^; (+) HRESIMS *m/z* 729.2018 [M + Na]^+^ (calcd for C_33_H_38_O_17_Na, 729.2001); ^1^H and ^13^C NMR data, see Table 4 and Table 5. All significant data are presented in Appendix A.

Phaneroside M (**13**): white amorphous powder; [α]${}_{D}^{20}$ + 22 (*c* 0.05, MeOH); UV (MeOH) *λ*_max_ (log *ε*) 217 (4.04), 319 (4.32) nm; IR (KBr) *ν*_max_ 3436, 1695, 1632, 1605, 1516, 1268, 1170, 1053 cm^−1^; (+) HRESIMS *m/z* 729.2010 [M + Na]^+^ (calcd for C_33_H_38_O_17_Na, 729.2001); ^1^H and ^13^C NMR data, see Table 4 and Table 5. All significant data are presented in Appendix A.

Phaneroside N (**14**): white amorphous powder; [α]${}_{D}^{20}$ + 28 (*c* 0.06, MeOH); UV (MeOH) *λ*_max_ (log *ε*) 232 (4.03), 319 (4.32) nm; IR (KBr) *ν*_max_ 3432, 1692, 1606, 1518, 1172, 1051 cm^−1^; (+) HRESIMS *m/z* 729.1990 [M + Na]^+^ (calcd for C_33_H_38_O_17_Na, 729.2001); ^1^H and ^13^C NMR data, see Table 4 and Table 5. All significant data are presented in Appendix A.

Phaneroside O (**15**): white amorphous powder; [α]${}_{D}^{20}$ + 22 (*c* 0.06, MeOH); UV (MeOH) *λ*_max_ (log *ε*) 217 (4.13), 237 (4.06), 327 (4.34) nm; IR (KBr) *ν*_max_ 3429, 1695, 1632, 1602, 1516, 1273, 1163, 1051 cm^−1^; (+) HRESIMS *m/z* 759.2124 [M + Na]^+^ (calcd for C_34_H_40_O_18_Na, 759.2107); ^1^H and ^13^C NMR data, see Table 4 and Table 5. All significant data are presented in Appendix A.

Phaneroside P (**16**): white amorphous powder; [α]${}_{D}^{20}$ + 27 (*c* 0.05, MeOH); UV (MeOH) *λ*_max_ (log *ε*) 217 (4.22), 236 (4.15), 325 (4.37) nm; IR (KBr) *ν*_max_ 3432, 1690, 1605, 1517, 1169, 1056 cm^−1^; (+) HRESIMS *m/z* 759.2095 [M + Na]^+^ (calcd for C_34_H_40_O_18_Na, 759.2107); ^1^H and ^13^C NMR data, see Table 4 and Table 5. All significant data are presented in Appendix A.

Phaneroside Q (**17**): white amorphous powder; [α]${}_{D}^{20}$ + 28 (*c* 0.06, MeOH); UV (MeOH) *λ*_max_ (log *ε*) 217 (4.07), 237 (4.01), 327 (4.29) nm; IR (KBr) *ν*_max_ 3426, 1697, 1632, 1598, 1516, 1272, 1161 cm^−1^; (+) HRESIMS *m/z* 759.2097 [M + Na]^+^ (calcd for C_34_H_40_O_18_Na, 759.2107); ^1^H and ^13^C NMR data, see Table 4 and Table 5. All significant data are presented in Appendix A.

Phaneroside R (**18**): white amorphous powder; [α]${}_{D}^{20}$ + 27 (*c* 0.06, MeOH); UV (MeOH) *λ*_max_ (log *ε*) 217 (4.13), 234 (4.04), 324 (4.20) nm; IR (KBr) *ν*_max_ 3428, 1696, 1605, 1517, 1261, 1170, 1054 cm^−1^; (+) HRESIMS *m/z* 759.2096 [M + Na]^+^ (calcd for C_34_H_40_O_18_Na, 759.2107); ^1^H and ^13^C NMR data, see Table 4 and Table 5. All significant data are presented in Appendix A.

Phaneroside S (**19**): white amorphous powder; [α]${}_{D}^{20}$ + 23 (*c* 0.06, MeOH); UV (MeOH) *λ*_max_ (log *ε*) 217 (4.17), 235 (4.07), 325 (4.26) nm; IR (KBr) *ν*_max_ 3397, 2921, 2850, 1646, 1516, 1272 cm^−1^; (+) HRESIMS *m/z* 759.2097 [M + Na]^+^ (calcd for C_34_H_40_O_18_Na, 759.2107); ^1^H and ^13^C NMR data, see Table 4 and Table 5. All significant data are presented in Appendix A.

Phaneroside T (**20**): white amorphous powder; [α]${}_{D}^{20}$ + 25 (*c* 0.06, MeOH); UV (MeOH) *λ*_max_ (log *ε*) 211 (3.92), 229 (3.97), 314 (4.32) nm; IR (KBr) *ν*_max_ 3420, 1697, 1606, 1516, 1262, 1171, 1056 cm^−1^; (+) HRESIMS *m/z* 741.2001 [M + Na]^+^ (calcd for C_34_H_38_O_17_Na 741.2001); ^1^H and ^13^C NMR data, see Table 6 and Table 7. All significant data are presented in Appendix A.

Phaneroside U (**21**): white amorphous powder; [α]${}_{D}^{20}$ + 26 (*c* 0.06, MeOH); UV (MeOH) *λ*_max_ (log *ε*) 211 (3.89), 228 (3.94), 314 (4.28) nm; IR (KBr) *ν*_max_ 3432, 1695, 1606, 1516, 1170, 1061 cm^−1^; (+) HRESIMS *m/z* 741.2001 [M + Na]^+^ (calcd for C_34_H_38_O_17_Na, 741.2001); ^1^H and ^13^C NMR data, see Table 6 and Table 7. All significant data are presented in Appendix A.

Phaneroside V (**22**): white amorphous powder; [α]${}_{D}^{20}$ + 26 (*c* 0.06, MeOH); UV (MeOH) *λ*_max_ (log *ε*) 230 (4.02), 316 (4.28) nm; IR (KBr) *ν*_max_ 3429, 1697, 1606, 1516, 1269, 1170, 1054 cm^−1^; (+) HRESIMS *m/z* 771.2093 [M + Na]^+^ (calcd for C_35_H_40_O_18_Na, 771.2107); ^1^H and ^13^C NMR data, see Table 6 and Table 7. All significant data are presented in Appendix A.

Phaneroside W (**23**): white amorphous powder; [α]${}_{D}^{20}$ + 23 (*c* 0.06, MeOH); UV (MeOH) *λ*_max_ (log *ε*) 217 (4.04), 319 (4.33) nm; IR (KBr) *ν*_max_ 3436, 1696, 1633, 1605, 1516, 1268, 1170, 1053 cm^−1^; (+) HRESIMS *m/z* 771.2112 [M + Na]^+^ (calcd for C_35_H_40_O_18_Na, 771.2107); ^1^H and ^13^C NMR data, see Table 6 and Table 7. All significant data are presented in Appendix A.

Phaneroside X (**24**): white amorphous powder; [α]${}_{D}^{20}$ + 25 (*c* 0.06, MeOH); UV (MeOH) *λ*_max_ (log *ε*) 217 (4.16), 238 (4.10), 328 (4.37) nm; IR (KBr) *ν*_max_ 3436, 1712, 1633, 1601, 1515, 1273, 1177 cm^−1^; (+) HRESIMS *m/z* 801.2226 [M + Na]^+^ (calcd for C_36_H_42_O_19_Na, 801.2213); ^1^H and ^13^C NMR data, see Table 6 and Table 7. All significant data are presented in Appendix A.

d.Acid Hydrolysis of Compound **1**

Compound **1** (4.0 mg) was added to 5.0 mL of 9% aqueous HCl in a sealed flask, which was refluxed at 80 °C for 5 h. The acidic aqueous mixture was dried, H_2_O (2 mL) was added, and the mixture was extracted with EtOAc (3 × 2 mL). The aqueous layer was concentrated to obtain the sugar fraction, which was dissolved with MeOH and analyzed using chiral-phase HPLC equipped with a Daicel Chiralpak AD-H column (250 × 4.6 mm, 5 μm) and an evaporative light-scattering detector (ELSD) using *n*-hexane:EtOH (82:18) as the mobile phase (0.7 mL/min) [27]. The sugars were confirmed to be D-glucose and D-fructose by comparing their retention times with those of D-glucose (17.4 min), L-glucose (18.2 min), D-fructose (25.6 min), and L-fructose (26.4 min) (Appendix A).

e.NO Production Measurements and Cell Viability Assays

The inhibitory effects of the isolated compounds on LPS-stimulated NO production were evaluated using the Griess reaction, and the cytotoxicities of compounds on BV-2 microglial cells were evaluated using MTT assays, as described in our previous report [23]. The result is shown in Figure 7.

f.Antioxidant Activity Assay

The antioxidant activity of the isolated compounds was tested using a DPPH radical scavenging assay as previously described, and vitamin C was used as the positive control [24].

## 4. Conclusions

In conclusion, 24 new phenylpropanoid esters of sucrose were isolated from the rattans of *P. championii*. Their structures were determined via extensive spectroscopic methods. The configuration of sugar moiety was determined via chiral-phase HPLC equipped with an evaporative light-scattering detector (ELSD) after acid hydrolysis of compound **1**. This is the first report of phenylpropanoid esters of sucrose isolated from the family Fabaceae. Structurally, these compounds revealed a huge structural diversity in terms of the number and position of phenylpropanoid and acetyl substituents. Biologically, all the isolated compounds were evaluated for their anti-inflammatory and antioxidant activities, and several compounds showed potent or moderate effects. Additionally, the structure–activity relationship was briefly discussed. These compounds may serve as potential leads for the development of anti-inflammatory agents.

## Figures and Tables

**Figure 1 molecules-28-04767-f001:**
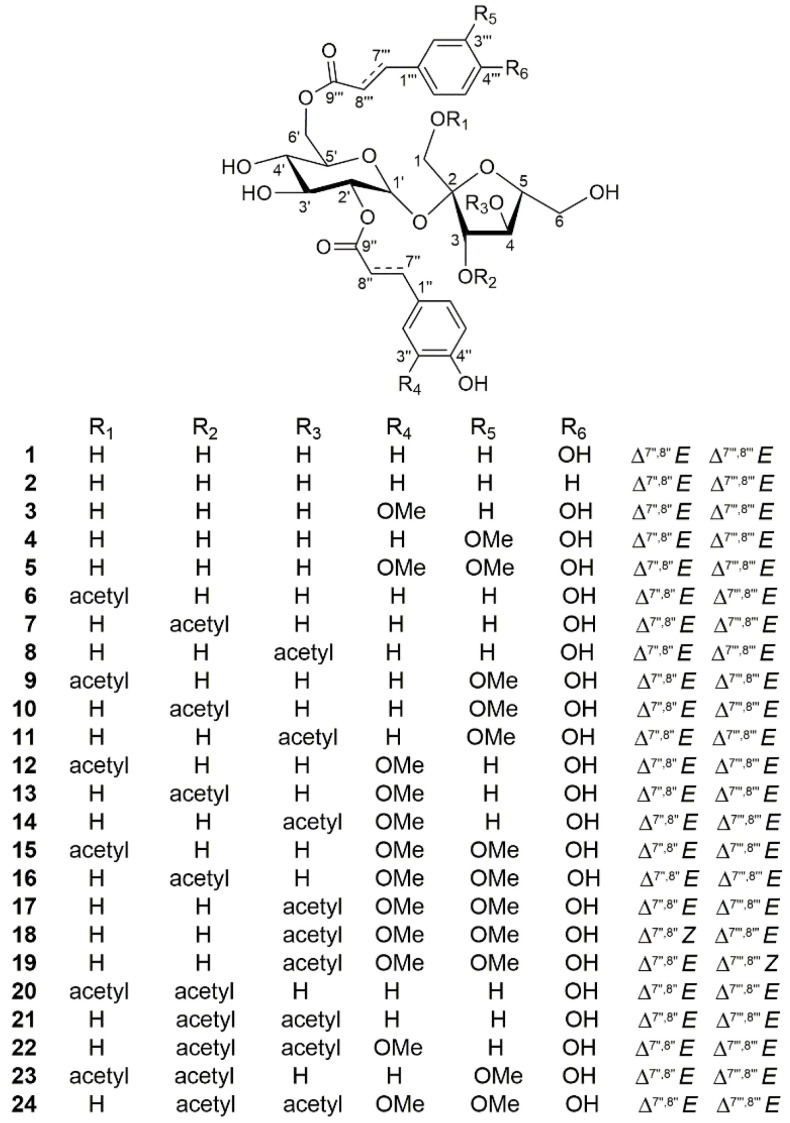
Chemical structures of compounds **1**–**24**.

**Figure 2 molecules-28-04767-f002:**
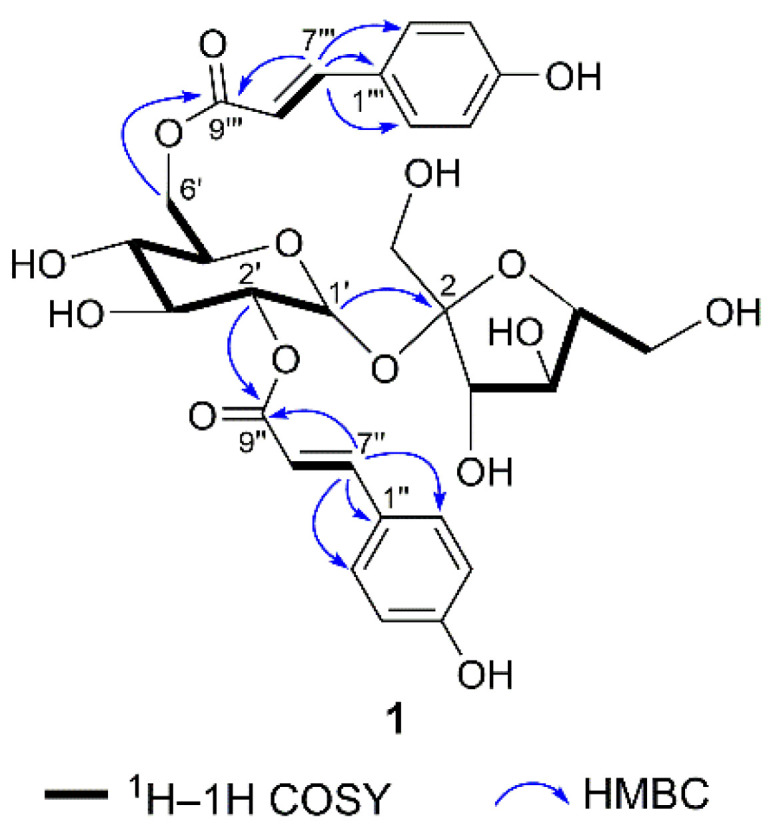
Key ^1^H-^1^H COSY and HMBC correlations of **1**.

**Figure 3 molecules-28-04767-f003:**
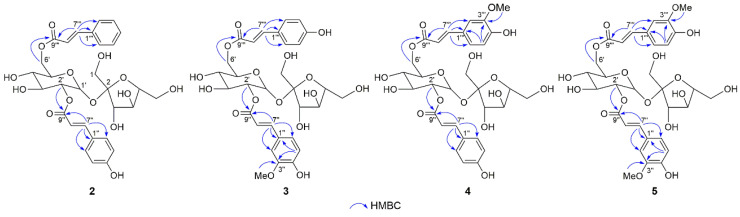
Key HMBC correlations of **2**–**5**.

**Figure 4 molecules-28-04767-f004:**
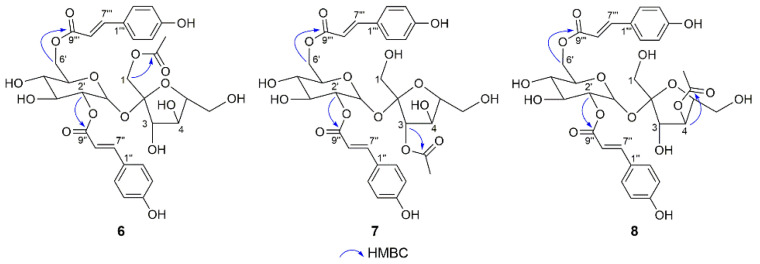
Key HMBC correlations of **6**, **7**, and **8**.

**Figure 5 molecules-28-04767-f005:**
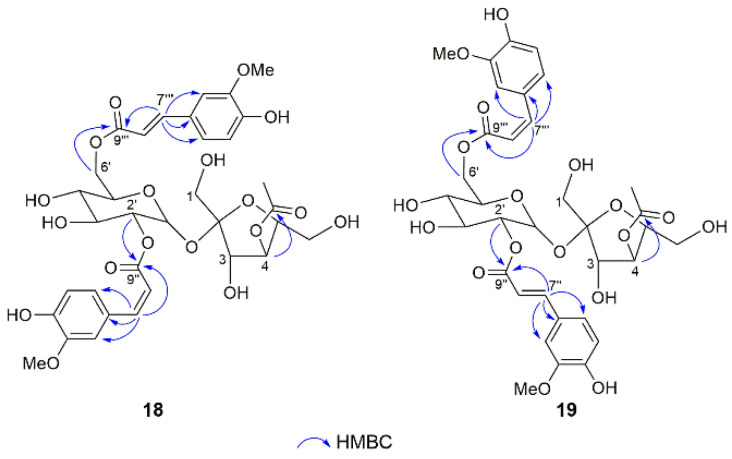
Key HMBC correlations of **18** and **19**.

**Figure 6 molecules-28-04767-f006:**
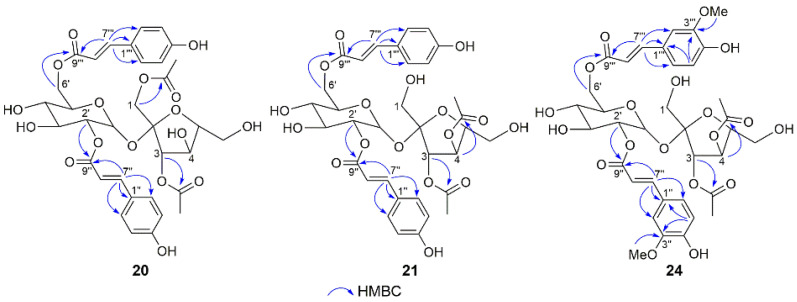
Key HMBC correlations of **20**, **21**, and **24**.

**Figure 7 molecules-28-04767-f007:**
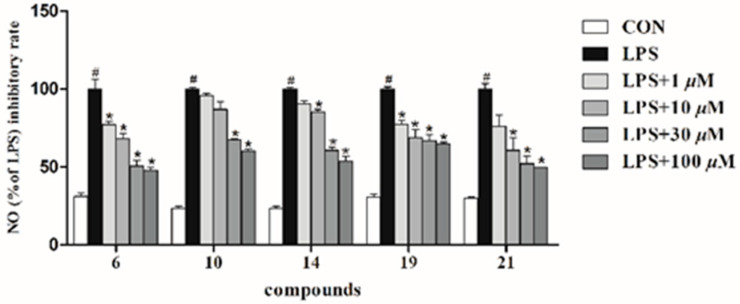
Effects of compounds **6**, **10**, **14**, **19**, and **21** on NO inhibitory activity (1–100 μM) in BV-2 microglial cells. * *p* < 0.001, compared with the LPS-treated group; # *p* < 0.001, compared with the control group.

**Table 1 molecules-28-04767-t001:** ^1^H NMR data of compounds **1**–**5** in *δ* ppm (*J* coupling in Hertz).

Position	1 ^a^	2 ^b^	3 ^a^	4 ^a^	5 ^a^
1	3.51, d (12.0) 3.31, d (12.0)	3.51, d (12.0) 3.31, overlapped	3.52, d (12.0) 3.32, d (12.0)	3.51, d (12.0) 3.30, d (12.0)	3.51, d (11.6) 3.31, overlapped
3	4.22, d (8.8)	4.22, d (9.0)	4.22, d (8.8)	4.22, d (8.4)	4.22, d (8.8)
4	4.04, t (8.8)	4.04, t (9.0)	4.04, t (8.8)	4.07, dd (9.2, 8.4)	4.07, t (8.8)
5	3.79, m	3.78, m	3.79, m	3.79, m	3.79, m
6	3.88, dd (12.0, 7.2) 3.77, m	3.88, dd (12.0, 7.2) 3.77, dd (12.0, 2.4)	3.88, m 3.77, m	3.88, m 3.77, m	3.89, overlapped 3.78, m
1′	5.60, d (3.6)	5.60, d (3.6)	5.60, d (4.0)	5.60, d (3.6)	5.61, d (3.6)
2′	4.73, dd (10.0, 3.6)	4.73, dd (9.6, 3.6)	4.73, dd (10.0, 4.0)	4.73, dd (9.6, 3.6)	4.73, dd (9.6, 3.6)
3′	4.01, dd (10.0, 9.2)	4.01, t (9.6)	4.01, t (10.0)	4.01, t (9.6)	4.01, t (9.6)
4′	3.45, t (9.2)	3.46, t (9.6)	3.46, t (10.0)	3.45, t (9.6)	3.45, t (9.6)
5′	4.19, ddd (9.2, 6.0, 2.0)	4.20, m	4.18, dd (10.0, 6.0, 2.0)	4.20, m	4.20, m
6′	4.54, dd (12.0, 2.0) 4.31, dd (12.0, 6.0)	4.56, br d (12.0) 4.34, dd (12.0, 6.6)	4.54, dd (12.0, 2.0) 4.31, dd (12.0, 6.0)	4.54, d (12.0) 4.31, dd (12.0, 6.4)	4.54, br d (12.0) 4.31, dd (12.0, 6.4)
2″	7.49, d (8.8)	7.49, d (8.4)	7.21, d (2.0)	7.48, d (8.4)	7.21, br s
3″	6.82, d (8.8)	6.81, d (8.4)		6.81, d (8.4)	
5″	6.82, d (8.8)	6.81, d (8.4)	6.82, d (8.0)	6.81, d (8.4)	6.82, d (8.4)
6″	7.49, d (8.8)	7.49, d (8.4)	7.11, dd (8.0, 2.0)	7.48, d (8.4)	7.11, br d (8.4)
7″	7.72, d (16.0)	7.72, d (16.2)	7.72, d (16.0)	7.72, d (16.0)	7.72, d (16.0)
8″	6.39, d (16.0)	6.39, d (16.2)	6.42, d (16.0)	6.39, d (16.0)	6.42, d (16.0)
2‴	7.51, d (8.8)	7.65, m	7.50, d (8.4)	7.25, br s	7.25, br s
3‴	6.81, d (8.8)	7.41, overlapped	6.81, d (8.4)		
4‴		7.41, overlapped			
5‴	6.81, d (8.8)	7.41, overlapped	6.81, d (8.4)	6.81, d (8.4)	6.81, d (8.4)
6‴	7.51, d (8.8)	7.65, m	7.50, d (8.4)	7.10, br d (8.4)	7.11, br d (8.4)
7‴	7.67, d (16.0)	7.75, d (16.2)	7.67, d (16.0)	7.65, d (16.0)	7.66, d (16.0)
8‴	6.43, d (16.0)	6.64, d (16.2)	6.43, d (16.0)	6.46, d (16.0)	6.46, d (16.0)
3″-OMe			3.90, s		3.90, s
3‴-OMe				3.89, s	3.90, s

^a^ In MeOH-*d*_4_, ^1^H NMR at 400 MHz. ^b^ In MeOH-*d*_4_, ^1^H NMR at 600 MHz. s = singlet; d = doublet; t = triplet; m = multiplet; br = broad.

**Table 2 molecules-28-04767-t002:** ^13^C NMR data of compounds **1**–**5** in *δ* ppm.

Position	1 ^a^	2 ^b^	3 ^a^	4 ^a^	5 ^a^
1	62.9	62.9	62.9	62.9	63.0
2	105.7	105.7	105.7	105.6	105.7
3	77.0	77.0	77.0	77.0	77.0
4	75.6	75.6	75.6	75.7	75.7
5	84.0	84.0	84.0	84.0	84.0
6	64.2	64.2	64.2	64.2	64.2
1′	90.6	90.6	90.6	90.5	90.5
2′	74.4	74.4	74.4	74.4	74.4
3′	71.9	71.9	71.9	72.0	72.0
4′	72.0	72.0	72.0	72.0	72.0
5′	71.9	71.9	71.9	71.9	71.9
6′	65.0	65.2	65.0	65.0	65.1
1″	127.1	127.0	127.6	127.0	127.7
2″	131.3	131.3	111.8	131.3	111.8
3″	116.9	116.9	149.4	116.9	149.4
4″	161.6	161.7	151.0	161.6	151.0
5″	116.9	116.9	116.5	116.9	116.5
6″	131.3	131.3	124.3	131.3	124.3
7″	147.5	147.5	147.8	147.5	147.8
8″	114.7	114.7	114.9	114.7	115.0
9″	168.8	168.8	168.8	168.8	168.8
1‴	127.0	135.8	127.1	127.7	127.6
2‴	131.3	129.4	131.3	111.6	111.6
3‴	116.8	130.0	116.8	149.4	149.4
4‴	161.4	131.6	161.4	150.7	150.8
5‴	116.8	130.0	116.8	116.4	116.4
6‴	131.3	129.4	131.3	124.3	124.3
7‴	146.9	146.7	146.9	147.1	147.1
8‴	114.9	118.7	115.0	115.2	115.2
9‴	169.2	168.5	169.2	169.1	169.1
3″-OMe			56.4		56.5
3‴-OMe				56.5	56.5

^a^ In MeOH-*d*_4_, ^13^C NMR at 100 MHz. ^b^ In MeOH-*d*_4_, ^13^C NMR at 150 MHz.

**Table 3 molecules-28-04767-t003:** ^1^H NMR data (400 MHz) of compounds **6**–**11** in MeOH-*d*_4_.

Position	6	7	8	9	10	11
1	4.11, d (11.6) 4.06, d (11.6)	3.54, d (12.0) 3.35, d (12.0)	3.57, d (12.0) 3.40, d (12.0)	4.11, d (12.0) 4.06, d (12.0)	3.54, d (12.0) 3.34, d (12.0)	3.57, br d (12.0) 3.40, br d (12.0)
3	4.08, d (8.8)	5.43, d (8.4)	4.47, d (8.4)	4.08, d (6.8)	5.44, d (8.4)	4.47, d (8.4)
4	4.04, t (8.8)	4.35, t (8.4)	5.30, t (8.4)	4.07, t (6.8)	4.39, t (8.4)	5.28, dd (8.4, 7.6)
5	3.79, m	3.90, m	3.91, m	3.79, m	3.91, m	3.91, overlapped
6	3.88, dd (11.6, 7.2) 3.75, m	3.84, dd (12.0, 6.4) 3.78, dd (12.0, 3.2)	3.92, m 3.77, dd (11.2, 2.4)	3.89, overlapped 3.76, m	3.85, dd (11.6, 6.8) 3.78, dd (11.6, 2.8)	3.93, m 3.80, dd (11.2, 2.8)
1-OAc	2.01, s			2.01, s		
3-OAc		2.18, s			2.18, s	
4-OAc			2.05, s			2.04, s
1′	5.65, d (3.6)	5.63, d (3.6)	5.65, d (3.6)	5.66, d (3.6)	5.63, d (3.6)	5.65, d (3.6)
2′	4.79, dd (10.4, 3.6)	4.72, dd (10.0, 3.6)	4.76, dd (10.0, 3.6)	4.80, dd (10.0, 3.6)	4.73, dd (10.4, 3.6)	4.76, dd (10.0, 3.6)
3′	4.00, dd (10.4, 9.2)	3.88, t (10.0)	4.01, dd (10.0, 9.6)	4.00, dd (10.0, 8.8)	3.88, overlapped	4.01, t (10.0)
4′	3.46, dd (10.0, 9.2)	3.47, t (10.0)	3.43, t (9.6)	3.45, dd (10.0, 8.8)	3.45, dd (10.0, 8.8)	3.44, t (10.0)
5′	4.20, ddd (10.0, 6.0, 2.0)	4.16, ddd (10.0, 6.0, 2.0)	4.23, m	4.21, ddd (10.0, 6.8, 2.0)	4.19, ddd (10.0, 6.8, 2.0)	4.24, ddd (10.0, 6.4, 1.6)
6′	4.54, dd (12.0, 2.0) 4.31, dd (12.0, 6.0)	4.56, dd (12.0, 2.0) 4.32, dd (12.0, 6.0)	4.57, dd (12.0, 1.6) 4.31, dd (12.0, 6.4)	4.54, dd (12.0, 2.0) 4.30, dd (12.0, 6.8)	4.56, dd (12.0, 2.0) 4.31, dd (12.0, 6.8)	4.57, dd (12.0, 1.6) 4.32, dd (12.0, 6.4)
2″	7.49, d (8.8)	7.48, d (8.8)	7.49, d (8.4)	7.49, d (8.8)	7.48, d (8.8)	7.49, d (8.8)
3″	6.80, d (8.8)	6.82, d (8.8)	6.81, d (8.4)	6.81, d (8.8)	6.81, d (8.8)	6.81, d (8.8)
5″	6.80, d (8.8)	6.82, d (8.8)	6.81, d (8.4)	6.81, d (8.8)	6.81, d (8.8)	6.81, d (8.8)
6″	7.49, d (8.8)	7.48, d (8.8)	7.49, d (8.4)	7.49, d (8.8)	7.48, d (8.8)	7.49, d (8.8)
7″	7.71, d (16.0)	7.70, d (16.0)	7.70, d (16.0)	7.71, d (16.0)	7.69, d (16.0)	7.70, d (16.0)
8″	6.39, d (16.0)	6.37, d (16.0)	6.40, d (16.0)	6.41, d (16.0)	6.37, d (16.0)	6.41, d (16.0)
2‴	7.50, d (8.8)	7.50, d (8.8)	7.51, d (8.4)	7.26, d (2.0)	7.25, d (2.0)	7.25, d (2.0)
3‴	6.80, d (8.8)	6.80, d (8.8)	6.80, d (8.4)			
5‴	6.80, d (8.8)	6.80, d (8.8)	6.80, d (8.4)	6.81, d (8.0)	6.81, d (8.0)	6.81, d (8.4)
6‴	7.50, d (8.8)	7.50, d (8.8)	7.51, d (8.4)	7.11, dd (8.0, 2.0)	7.09, dd (8.0, 2.0)	7.11, dd (8.4, 2.0)
7‴	7.66, d (16.0)	7.66, d (16.0)	7.65, d (16.0)	7.65, d (15.6)	7.65, d (16.0)	7.65, d (16.0)
8‴	6.43, d (16.0)	6.43, d (16.0)	6.49, d (16.0)	6.47, d (15.6)	6.47, d (16.0)	6.52, d (16.0)
3″-OMe						3.90, s
3‴-OMe				3.90, s	3.89, s	

s = singlet; d = doublet; t = triplet; m = multiplet; br = broad.

**Table 4 molecules-28-04767-t004:** ^1^H NMR data of compounds **12**–**19**.

Position	12 ^a^	13 ^a^	14 ^a^	15 ^a^	16 ^b^	17 ^a^	18 ^b^	19 ^b^
1	4.14, d (12.4) 4.03, d (12.4)	3.55, d (11.6) 3.36, d (11.6)	3.59, d (12.0) 3.42, d (12.0)	4.14, d (11.6) 4.02, d (11.6)	3.54, d (12.0) 3.34, d (12.0)	3.59, d (12.0) 3.42, d (12.0)	3.55, d (12.0) 3.39, d (12.0)	3.58, d (12.0) 3.42, d (12.0)
3	4.09, d (8.4)	5.43, d (8.4)	4.47, d (8.0)	4.09, d (8.4)	5.44, d (8.4)	4.48, d (8.8)	4.47, d (8.4)	4.46, d (8.4)
4	4.04, t (8.4)	4.35, t (8.4)	5.31, t (8.0)	4.08, t (8.4)	4.39, t (8.4)	5.29, dd (8.8, 7.6)	5.26, t (8.4)	5.29, t (8.4)
5	3.79, m	3.90, overlapped	3.91, overlapped	3.80, m	3.91, overlapped	3.91, overlapped	3.90, overlapped	3.90, overlapped
6	3.88, m 3.75, dd (11.6, 2.8)	3.84, dd (12.0, 6.4) 3.78, dd (12.0, 3.2)	3.92, overlapped 3.78, m	3.89, overlapped 3.76, dd (11.6, 2.8)	3.85, dd (12.0, 6.6) 3.79, dd (12.0, 2.4)	3.92, overlapped 3.81, dd (11.2, 2.4)	3.92, overlapped 3.78, dd (10.8, 2.4)	3.90, overlapped 3.78, m
1-OAc	2.02, s			2.02, s				
3-OAc		2.18, s			2.18, s			
4-OAc			2.04, s			2.03, s	2.08, s	2.05, s
1′	5.64, d (3.6)	5.63, d (4.0)	5.67, d (3.6)	5.65, d (4.0)	5.64, d (3.6)	5.66, d (4.0)	5.62, d (3.6)	5.66, d (3.6)
2′	4.82, dd (10.4, 3.6)	4.72, dd (10.4, 4.0)	4.76, dd (10.0, 3.6)	4.82, dd (10.0, 4.0)	4.72, dd (10.2, 3.6)	4.76, dd (10.0, 4.0)	4.75, dd (10.2, 3.6)	4.73, dd (10.2, 3.6)
3′	4.00, dd (10.4, 9.2)	3.88, overlapped	4.01, t (10.0)	4.00, dd (10.0, 8.8)	3.87, t (10.2)	4.01, dd (10.0, 8.8)	3.98, dd (10.2, 9.6)	3.99, dd (10.2, 9.0)
4′	3.46, dd (10.0, 9.2)	3.47, t (10.0)	3.43, t (10.0)	3.46, dd (10.0, 8.8)	3.45, t (10.2)	3.44, dd (10.0, 8.8)	3.43, t (9.6)	3.43, dd (10.2, 9.0)
5′	4.20, ddd (10.0, 6.4, 2.0)	4.16, ddd (10.0, 6.0, 2.0)	4.23, dd (10.0, 6.4)	4.22, ddd (10.0, 6.4, 2.0)	4.19, m	4.23, ddd (10.0, 6.4, 1.6)	4.23, ddd (9.6, 6.6, 1.8)	4.19, ddd (10.2, 5.4, 1.8)
6′	4.54, dd (12.0, 2.0) 4.31, dd (12.0, 6.4)	4.56, dd (12.0, 2.0) 4.32, dd (12.0, 6.0)	4.57, br d (12.0) 4.31, dd (12.0, 6.4)	4.54, dd (12.0, 2.0) 4.30, dd (12.0, 6.4)	4.56, br d (12.0) 4.31, dd (12.0, 6.6)	4.57, dd (12.0, 1.6) 4.32, dd (12.0, 6.4)	4.56, dd (12.0, 1.8) 4.31, dd (12.0, 6.6)	4.53, dd (12.0, 1.8) 4.30, dd (12.0, 5.4)
2″	7.26, d (2.0)	7.21, d (2.0)	7.23, d (2.0)	7.26, d (2.4)	7.21, br s	7.23, d (2.0)	7.93, d (2.4)	7.23, d (1.8)
5″	6.81, d (8.4)	6.82, d (8.0)	6.82, d (8.0)	6.81, d (8.0)	6.82, d (7.8)	6.81, d (8.0)	6.77, d (8.4)	6.81, d (8.4)
6″	7.09, dd (8.4, 2.0)	7.10, dd (8.0, 2.0)	7.09, dd (8.0, 2.0)	7.09, dd (8.0, 2.4)	7.10, overlapped	7.09, dd (8.0, 2.0)	7.19, dd (8.4, 2.4)	7.09, dd (8.4, 1.8)
7″	7.70, d (15.6)	7.69, d (15.6)	7.69, d (16.0)	7.70, d (16.0)	7.69, d (16.2)	7.69, d (16.0)	6.92, d (12.6)	7.68, d (15.6)
8″	6.45, d (15.6)	6.39, d (15.6)	6.44, d (16.0)	6.45, d (16.0)	6.40, d (16.2)	6.44, d (16.0)	5.88, d (12.6)	6.44, d (15.6)
2‴	7.51, d (8.4)	7.50, d (8.8)	7.51, d (8.0)	7.26, d (2.4)	7.26, br s	7.24, d (2.0)	7.25, d (2.4)	7.87, d (1.8)
3‴	6.81, d (8.4)	6.80, d (8.8)	6.80, d (8.0)					
5‴	6.81, d (8.4)	6.80, d (8.8)	6.80, d (8.0)	6.81, d (8.0)	6.80, d (7.8)	6.81, d (8.0)	6.81, d (8.4)	6.77, d (8.4)
6‴	7.51, d (8.4)	7.50, d (8.8)	7.51, d (8.0)	7.11, dd (8.0, 2.4)	7.10, overlapped	7.11, dd (8.0, 2.0)	7.11, dd (8.4, 2.4)	7.15, dd (8.4, 1.8)
7‴	7.66, d (15.6)	7.66, d (15.6)	7.65, d (16.0)	7.65, d (16.0)	7.65, d (16.2)	7.65, d (16.0)	7.65, d (16.2)	6.88, d (13.2)
8‴	6.43, d (15.6)	6.43, d (15.6)	6.49, d (16.0)	6.47, d (16.0)	6.47, d (16.2)	6.52, d (16.0)	6.53, d (16.2)	5.90, d (13.2)
3″-OMe	3.92, s	3.90, s	3.92, s	3.92, s	3.90, s	3.92, s	3.88, s	3.92, s
3‴-OMe				3.90, s	3.89, s	3.90, s	3.90, s	3.89, s

^a^ In MeOH-*d*_4_, ^1^H NMR at 400 MHz. ^b^ In MeOH-*d*_4_, ^1^H NMR at 600 MHz. s = singlet; d = doublet; t = triplet; m = multiplet; br = broad.

**Table 5 molecules-28-04767-t005:** ^13^C NMR data of compounds **12**–**19**.

Position	6 ^a^	7 ^a^	8 ^a^	9 ^a^	10 ^a^	11 ^a^	12 ^a^	13 ^a^	14 ^a^	15 ^a^	16 ^b^	17 ^a^	18 ^b^	19 ^b^
1	64.8	64.4	62.5	64.8	64.5	62.5	64.8	64.5	62.5	64.8	64.5	62.5	62.5	62.3
2	103.9	105.0	106.0	103.8	104.9	106.0	103.9	105.0	106.1	103.8	104.9	106.1	105.9	106.2
3	78.5	78.6	75.2	78.5	78.6	75.2	78.4	78.7	75.2	78.4	78.6	75.2	75.2	75.2
4	75.3	73.8	78.0	75.4	73.8	78.1	75.4	73.8	78.0	75.4	73.8	78.2	78.2	77.9
5	84.1	84.2	82.2	84.1	84.2	82.3	84.2	84.2	82.3	84.2	84.2	82.3	82.3	82.3
6	64.1	63.9	64.8	64.2	64.0	64.8	64.2	63.9	64.9	64.3	64.0	64.9	64.8	64.8
1-OAc	172.0 20.6			172.0 20.6			172.0 20.6			172.0 20.6				
3-OAc		172.2 20.8			172.2 20.8			172.2 20.8			172.2 20.8			
4-OAc			172.4 20.8			172.4 20.8			172.3 20.8			172.4 20.8	172.4 20.9	172.3 20.8
1′	90.9	90.5	91.1	90.9	90.4	91.1	91.0	90.6	91.2	90.9	90.4	91.2	91.0	91.3
2′	74.1	74.3	74.2	74.1	74.3	74.3	74.1	74.3	74.2	74.1	74.3	74.2	73.9	74.2
3′	72.1	72.3	72.0	72.2	72.3	72.0	72.2	72.3	72.0	72.2	72.3	72.1	71.9	72.0
4′	72.0	71.8	72.0	72.0	71.9	72.1	72.1	72.0	72.1	72.0	71.9	72.0	72.1	71.9
5′	72.0	72.0	72.0	72.0	72.0	72.0	72.0	71.8	72.0	72.0	72.0	72.0	72.0	71.8
6′	65.0	64.9	65.1	65.1	65.1	65.1	65.0	64.9	65.1	65.1	65.1	65.1	65.1	64.5
1″	127.1	127.1	127.0	127.2	127.0	127.1	127.7	127.5	127.7	127.8	127.5	127.6	127.9	127.5
2″	131.3	131.3	131.3	131.4	131.3	131.4	111.7	111.8	111.7	111.6	111.8	111.8	115.3	111.7
3″	116.9	116.8	116.9	116.8	116.9	116.8	149.4	149.5	149.4	149.4	149.5	149.4	148.3	149.5
4″	161.7	161.6	161.6	161.4	161.6	161.4	150.8	151.5	150.8	150.8	151.1	150.9	149.9	150.9
5″	116.9	116.8	116.9	116.8	116.9	116.8	116.4	116.6	116.4	116.4	116.5	116.4	115.6	116.5
6″	131.3	131.3	131.3	131.4	131.3	131.4	124.5	124.3	124.4	124.5	124.3	124.4	127.4	124.5
7″	147.5	147.5	147.4	147.5	147.5	147.3	147.7	147.8	147.5	147.7	147.8	147.6	147.0	147.5
8″	114.8	114.6	114.9	114.9	114.6	114.9	115.2	114.9	115.2	115.2	114.9	115.2	115.8	115.1
9″	168.8	168.7	168.8	168.8	168.7	168.8	168.7	168.7	168.8	168.7	168.7	168.8	167.5	168.8
1‴	127.1	127.0	127.2	127.7	127.6	127.8	127.2	127.1	127.3	127.7	127.6	127.8	127.7	128.0
2‴	131.4	131.3	131.4	111.6	111.5	111.9	131.3	131.3	131.3	111.7	111.5	111.7	111.8	115.1
3‴	116.9	116.9	116.8	149.4	149.4	149.4	116.8	116.9	116.8	149.4	149.5	149.4	149.4	148.4
4‴	161.6	161.4	161.4	150.7	150.8	150.6	161.4	161.5	161.3	150.7	150.9	150.7	150.8	149.7
5‴	116.9	116.9	116.8	116.4	116.4	116.4	116.8	116.9	116.8	116.4	116.4	116.5	116.4	115.7
6‴	131.4	131.3	131.4	124.4	124.4	124.4	131.3	131.3	131.3	124.4	124.5	124.4	124.4	127.0
7‴	146.9	146.9	146.8	147.1	147.1	147.1	146.9	146.9	146.8	147.1	147.2	147.1	147.1	146.0
8‴	114.9	114.9	115.2	115.3	115.2	115.4	115.0	114.9	115.3	115.3	115.2	115.4	115.4	116.1
9‴	169.2	169.2	169.3	169.1	169.1	169.3	169.2	169.2	169.3	169.1	169.1	169.3	169.3	168.1
3″-OMe						56.5	56.5	56.4	56.5	56.5	56.5	56.5	56.4	56.4
3‴-OMe				56.5	56.5					56.5	56.4	56.5	56.5	56.4

^a^ In MeOH-*d*_4_, ^13^C NMR at 100 MHz. ^b^ In MeOH-*d*_4_, ^13^C NMR at 150 MHz.

**Table 6 molecules-28-04767-t006:** ^1^H NMR data of compounds **20**–**24**.

Position	20 ^a^	21 ^a^	22 ^b^	23 ^a^	24 ^a^
1	4.17, d (12.0) 4.00, d (12.0)	3.57, d (12.0) 3.45, d (12.0)	3.58, d (12.0) 3.47, d (12.0)	4.16, d (11.6) 3.99, d (11.6)	3.59, d (12.0) 3.48, d (12.0)
3	5.29, d (8.0)	5.65, d (7.6)	5.65, d (7.8)	5.30, d (8.4)	5.65, d (7.6)
4	4.34, t (8.0)	5.48, t (7.6)	5.50, t (7.8)	4.38, t (8.4)	5.49, t (7.6)
5	3.89, m	4.07, m	4.08, m	3.90, overlapped	4.08, td (7.6, 4.4)
6	3.84, dd (11.6, 6.8) 3.77, dd, (11.6, 2.8)	3.87, dd (12.0, 6.8) 3.77, dd (12.0, 4.4)	3.87, dd (12.0, 7.2) 3.78, dd (12.0, 4.2)	3.85, dd (12.0, 6.8) 3.77, dd (12.0, 2.8)	3.87, overlapped 3.80, dd (12.0, 4.4)
1-OAc	2.02, s			2.02, s	
3-OAc	2.17, s	2.11, s	1.98, s	2.18, s	2.11, s
4-OAc		1.99, s	2.10, s		1.97, s
1′	5.66, d (3.6)	5.62, d (3.6)	5.62, d (3.6)	5.66, d (3.6)	5.62, d (3.6)
2′	4.78, dd (10.0, 3.6)	4.76, dd (10.0, 3.6)	4.77, dd (10.2, 3.6)	4.78, dd (10.0, 3.6)	4.77, dd (10.0, 3.6)
3′	3.88, dd (10.0, 9.2)	3.91, dd (10.0, 9.2)	3.91, m	3.88, m	3.91, overlapped
4′	3.47, t (9.2)	3.47, t (9.2)	3.46, t (10.2)	3.46, dd (10.0, 9.2)	3.47, m
5′	4.16, m	4.19, ddd (10.0, 6.0, 2.0)	4.18, ddd (10.2, 6.0, 1.8)	4.18, m	4.19, ddd (10.0, 6.4, 2.0)
6′	4.57, dd (12.0, 2.0) 4.32, dd (12.0, 6.0)	4.60, dd (12.0, 2.0) 4.33, dd (12.0, 6.0)	4.59, dd (12.0, 1.8) 4.32, dd (12.0, 6.0)	4.57, dd (12.0, 2.0) 4.30, dd (12.0, 6.8)	4.60, dd (12.0, 2.0) 4.33, dd (12.0, 6.4)
2″	7.49, d (8.8)	7.49, d (8.8)	7.25, br s	7.49, d (8.8)	7.24, d (2.0)
3″	6.81, d (8.8)	6.81, d (8.8)		6.81, d (8.8)	
5″	6.81, d (8.8)	6.81, d (8.8)	6.81, d (8.4)	6.81, d (8.8)	6.82, d (8.0)
6″	7.49, d (8.8)	7.49, d (8.8)	7.10, dd (8.4, 1.8)	7.49, d (8.8)	7.10, dd (8.0, 2.0)
7″	7.68, d (15.6)	7.69, d (16.0)	7.69, d (16.2)	7.68, d (16.0)	7.69, d (16.0)
8″	6.38, d (15.6)	6.41, d (16.0)	6.45, d (16.2)	6.38, d (16.0)	6.45, d (16.0)
2‴	7.50, d (8.8)	7.50, d (8.8)	7.49, d (8.4)	7.26, d (2.0)	7.24, d (2.0)
3‴	6.81, d (8.8)	6.81, d (8.8)	6.80, d (8.4)		
5‴	6.81, d (8.8)	6.81, d (8.8)	6.80, d (8.4)	6.81, d (8.0)	6.82, d (8.0)
6‴	7.50, d (8.8)	7.50, d (8.8)	7.49, d (8.4)	7.09, dd (8.0, 2.0)	7.10, dd (8.0, 2.0)
7‴	7.66, d (16.0)	7.65, d (16.0)	7.65, d (16.2)	7.65, d (16.0)	7.65, d (16.0)
8‴	6.43, d (16.0)	6.45, d (16.0)	6.44, d (16.2)	6.47, d (16.0)	6.49, d (16.0)
3″-OMe			3.91, s		3.90, s
3‴-OMe				3.89, s	3.91, s

^a^ In MeOH-*d*_4_, ^1^H NMR at 400 MHz. ^b^ In MeOH-*d*_4_, ^1^H NMR at 600 MHz. s = singlet; d = doublet; t = triplet; m = multiplet; br = broad.

**Table 7 molecules-28-04767-t007:** ^13^C NMR data of compounds **20**–**24**.

Position	20 ^a^	21 ^a^	22 ^b^	23 ^a^	24 ^a^
1	66.1	63.6	63.4	66.2	63.5
2	103.3	106.1	106.2	103.2	106.2
3	79.5	76.6	76.6	79.5	76.7
4	73.5	76.5	76.5	73.5	76.6
5	84.2	82.7	82.7	84.3	82.7
6	63.8	64.1	64.2	63.9	64.3
1-OAc	172.0 20.6			171.9 20.6	
3-OAc	172.1 20.7	171.8 20.7	172.0 20.7	172.1 20.7	171.7 20.7
4-OAc		172.0 20.7	171.8 20.7		172.0 20.7
1′	91.0	91.3	91.4	90.9	91.4
2′	74.0	74.1	74.1	74.0	74.1
3′	72.3	72.2	72.2	72.4	72.2
4′	71.8	71.8	71.9	71.9	71.9
5′	72.2	72.2	72.2	72.2	72.2
6′	64.9	64.9	64.9	65.1	64.9
1″	127.1	127.2	127.4	127.1	127.6
2″	131.3	131.3	111.7	131.4	111.7
3″	116.8	116.8	149.5	116.8	149.4
4″	161.5	161.6	151.2	161.5	150.7
5″	116.8	116.8	116.5	116.8	116.4
6″	131.3	131.3	124.6	131.4	124.3
7″	147.5	147.5	147.8	147.5	147.8
8″	114.7	114.8	115.0	114.8	115.1
9″	168.7	168.7	168.7	168.7	168.7
1‴	127.1	127.0	127.1	127.7	127.8
2‴	131.4	131.4	131.3	111.5	111.8
3‴	116.8	116.9	116.9	149.4	149.4
4‴	161.4	161.4	161.6	150.7	150.8
5‴	116.8	116.9	116.9	116.4	116.4
6‴	131.4	131.4	131.3	124.5	124.5
7‴	146.9	146.8	146.9	147.1	147.1
8‴	114.9	115.1	115.0	115.3	115.4
9‴	169.2	169.2	169.2	169.1	169.2
3″-OMe			56.4		56.5
3‴-OMe				56.5	56.5

^a^ In MeOH-*d*_4_, ^13^C NMR at 100 MHz. ^b^ In MeOH-*d*_4_, ^13^C NMR at 150 MHz.

## Data Availability

The data of the article can be obtained from the authors.

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
