# Peer review of "Phanerosides A–X, Phenylpropanoid Esters of Sucrose from the Rattans of Phanera championii Benth"

_molecules, 2023, doi:10.3390/molecules28124767_

Round 1

Reviewer 1 Report

The authors report twenty-four new phenylpropanoid esters of sucrose from an EtOH extract of the rattans of Phanera championii Benth., along with their activities. This manuscript is well-prepared. However, there are still some minor mistakes that need to be revised.

- Check the order of the page numbers. They are not continuous.

- Abstract: Change "DPPH radical scavenging activities" to "DPPH radical scavenging activity."

- Figure 2: Check the COSY correlation between C-2 and C-3 because C-2 is a quaternary carbon.

- General Experimental Procedures: Delete the sentence "ECD spectra were measured using a JASCO J-1500 spectropolarimeter" if this experiment was not performed.

- Change "a. Physicochemical Properties and Spectroscopic Data of Compounds 1–2" to "c. Physicochemical Properties and Spectroscopic Data of Compounds 1–24".

- Change "b. Acid Hydrolysis of Compound 1" to "d. Acid Hydrolysis of Compound 1".

- Change "c. NO Production Measurements and Cell Viability Assays" to "e. NO Production Measurements and Cell Viability Assays."

- Change "d. Antioxidant activity assay" to "f. Antioxidant Activity Assay."

- Change "The antioxidant activities of the isolated compounds were tested by a DPPH radical scavenging assay" to "The antioxidant activity of the isolated compounds was tested by a DPPH radical scavenging assay."

There are still some minor grammar errors that need to be revisited.

Author Response

Comment 1:

Check the order of the page numbers. They are not continuous.

Response:

Sorry for our carelessness. The order of the page numbers has been corrected.

Comment 2:

Abstract: Change "DPPH radical scavenging activities" to "DPPH radical scavenging activity."

Response:

Thank you for your correction. "DPPH radical scavenging activities" in the “Abstract” part has been corrected to "DPPH radical scavenging activity" in the revised manuscript.

Comment 3:

Figure 2: Check the COSY correlation between C-2 and C-3 because C-2 is a quaternary carbon.

Response:

Thank you for your reminding. Figure 2 has been replaced with the reasonable COSY correlations.

Comment 4:

General Experimental Procedures: Delete the sentence "ECD spectra were measured using a JASCO J-1500 spectropolarimeter" if this experiment was not performed.

Response:

Sorry for our carelessness. The sentence "ECD spectra were measured using a JASCO J-1500 spectropolarimeter" in the “General Experimental Procedures” part has been deleted because the ECD data is not present in the manuscript.

Comment 5:

Change "a. Physicochemical Properties and Spectroscopic Data of Compounds 1–2" to "c. Physicochemical Properties and Spectroscopic Data of Compounds 1–24"

Response:

Thank you for your correction. We have made corresponding corrections in the manuscript.

Comments 6, 7, and 8:

Change "b. Acid Hydrolysis of Compound 1" to "d. Acid Hydrolysis of Compound 1"; Change "c. NO Production Measurements and Cell Viability Assays" to "e. NO Production Measurements and Cell Viability Assays."; Change "d. Antioxidant activity assay" to "f. Antioxidant Activity Assay."

Response:

Thank you for your correction. The order of the subheadings you point has been corrected and is continuous according to your comments.

Comment 9:

Change "The antioxidant activities of the isolated compounds were tested by a DPPH radical scavenging assay" to "The antioxidant activity of the isolated compounds was tested by a DPPH radical scavenging assay."

Response:

Sorry for our mistake. The sentence you point above has been modified according to your suggestion.

Reviewer 2 Report

The manuscript describes the isolation and structure elucidation of 24 phenylpropanoid esters of sucrose for the first time in family Fabaceae, in addition to their in-vitro antioxidant and anti-inflammatory effects. The manuscript is overall very well-written, has enough novelty and descriptive, so I recommend its publication after considering some comments:

1. line 44, clarify that the biology was carried out in-vitro in the aim of study statement. Also  change isolates to isolated compounds.

2. Line 238, I think the use of 100 L of solvent is not economical or environmentally friendly and makes it hard to be reproducible. So why you didn't use a smaller scale.

3. Since biological evaluation was carried out on all compounds, I suggest adding little discussion on SAR as I noticed for example that presence of methoxy group in the compound have decreased the anti-inflammatory effect observed to some extent.

Author Response

Comment 1:

line 44, clarify that the biology was carried out in-vitro in the aim of study statement. Also change isolates to isolated compounds.

Response:

Thank you for your kind suggestion and sorry for our lack of precision. We have explained in the revised manuscript that the anti-inflammatory and antioxidant activities tests of all compounds were carried out in vitro. And the “isolates” was changed to “isolated compounds” in the corresponding position of this manuscript.

Comment 2:

Line 238, I think the use of 100 L of solvent is not economical or environmentally friendly and makes it hard to be reproducible. So why you didn't use a smaller scale.

Response:

Thank you for your meaningful question, environmental protection is always under our consideration. There are two reasons for the amount of solvent. Firstly, the rattans of Phanera championii, as extraction object, have taken up so much space that they could be submerged just enough to achieve extraction conditions with 100 L of solvent. Then, the solvent was recycled after each extraction for environmentally friendly.

Comment 3:

Since biological evaluation was carried out on all compounds, I suggest adding little discussion on SAR as I noticed for example that presence of methoxy group in the compound have decreased the anti-inflammatory effect observed to some extent.

Response:

Thank you for your kind suggestion. I coincided with your idea during my writing. However, I found that I cannot draw an infallible conclusion. For the five compounds with anti-inflammatory activity, the activity of the ones with methoxy group in the structure was weaker than those without. But some compounds, like 1, 2, 7, and 8, showed no significant activity even without methoxy group in their structures. Therefore, I think that there are various factors determining whether a compound is active or not. The number and position of substituents, such as acetyl group and methoxy group, may all affect the effectiveness of their anti-inflammatory activity.

Reviewer 3 Report

This is a huge work of extraction, separation and characterization of new disubsituted phenyl propanoid sucroester. 

Please check line 13 the Fabacae family

line 43 removed and before anti-inflammatory

Table 1. 1H NMR Data of Compounds 1‒5 in ppm (J coupling in Hertz)

Table 2 add in d ppm

line 80 the desperately analogous ??

Line 115, 131, amu? mass unit more

It's OK few mistake.

Author Response

Comment 1:

Please check line 13 the Fabaceae family.

Response:

Thank you for your reminding. “the family Fabaceae” has been changed to “the Fabaceae family” in line 13 of the manuscript in the light of your suggestion.

Comment 2:

line 43 removed and before anti-inflammatory.

Response:

Thank you for your correction. The “and” before anti-inflammatory in line 43 has been deleted according to your suggestion.

Comment 3:

Table 1. 1H NMR Data of Compounds 1‒5 in δ ppm (J coupling in Hertz); Table 2 add in δ ppm.

Response:

Thank you for your kind suggestion. The titles of table 1 and table 2 have been completed in the revised manuscript.

Comment 4:

line 80 the desperately analogous

Response:

    We have made emendation in the corresponding place in line 80 of the manuscript.

Comment 5:

Line 115, 131, amu … mass unit more.

Response:

Thank you for your correction. The corresponding points have been modified.